# Modal Analysis of Bolted Structure Based on Equivalent Material of Joint Interface

**DOI:** 10.3390/ma12183004

**Published:** 2019-09-16

**Authors:** Kai Zhang, Guoxi Li, Jingzhong Gong, Fei Wan

**Affiliations:** 1College of Intelligence Science and Technology, National University of Defense Technology, No.109 Deya Street, Changsha 410073, China; lgx2020@sina.com (G.L.); feifeiwenqing123@163.com (F.W.); 2School of Mechanical Engineering, Hunan International Economics University, No.822 Fenglin Street, Changsha 410073, China; jzgong@nudt.edu.cn

**Keywords:** bolted joint, joint interface, equivalent material, modal analysis, finite element simulation

## Abstract

Modal performance of a bolted structure is important when considering a precise mechanical product. Joint interface is a critical aspect of a bolted structure, which is difficult to analyze because of its rough profiles. Equivalent material (EM) is used to simplify the joint interface, which reduces the computational cost and modeling difficulty. Using a modified fractal model based on an oblique asperity contact, we calculated the elastic modulus, the shear modulus, the Poisson’s ratio, and the density of EM. The finite element method was applied to discriminate between the resonant frequencies of bolted structures with EM and without EM. The simulation results are compared with the classical frequency response experiment. The errors between experiments and the bolted structures with EM are less than 10%, which are much less than those without EM. It can be concluded that bolted structures with EM are more reliable and reasonable. Furthermore, the effect of EM is more obvious when the joint interface has a small size, roughness, and tightening torque.

## 1. Introduction

It is challenging to analyze the dynamic characteristics of some structures and materials in engineering because of complexity, scale difference, and theoretical modeling difficulties of rough interfaces between parts. In this study, we propose equivalent material (EM) (i.e., virtual material or equivalent interval) to replace the rough interfaces of certain structures and materials. The properties of EM are calculated based on specific methods. In this way, the computational cost and modeling difficulty is reduced in the simulation and theoretical calculation while the computational cost is also promised. EM is widely used in machined rough joints [1], bolted structures [2], three-dimensional (3D) printings [3], fiber structures [4], open-cell foams, [5] electrical machine sets thermal analyses [6], biocompatible materials [7], and integrated circuits [8]. Most simulation results are in good agreement with the experimental results.

As an important mechanical coupling method, the bolted structure has an important influence on the dynamic performance of the precision mechanical structure [9], where the modal index is an important indicator. Finite element simulation is a commonly used performance prediction method. A large error would occur if the rough topography of the joint interfaces was not considered. It is complicated to generate the mesh of the rough topography in finite element simulation. Therefore, the EM method can be utilized to simplify the joint interface. The elastic modulus, Poisson’s ratio, and other properties can be calculated by the mature theories.

There are many methods for theoretical calculation of contact interfaces, such as the asperity-based theory (Greenwood-Williamson model (GW model) [10], the Bush-Gibson-Thomas model (BGT model) [11]), the fractal theory (Majumdar-Bushan model (MB model) [12], the Wang-Komvopoulos model (WK model) [13]), and Persson’s theory [14]. Many scholars continue to increase assumptions (e.g., asperity shape, multi-scale characteristic, elastic-plastic deformation) and expand the contact factors (e.g., friction, bonding, lubrication). These improvements and extensions make the theoretical model more accurate. Tian et al. [9] studied the dynamic characteristics of the fixed joint surface of the machine tool by using the isotropic virtual material. The Hertz contact theory and fractal theory are applied to calculate the properties. The error of modal response and resonant frequency between simulations and experimental results was less than 9%. Similarly, Zhao et al. [15] also performed virtual material analysis on the bolted structure of the machine tool by the weighted mean method. The uneven distribution of contact stress was discussed using the finite element method. The error of the results was less than 4.41%, which also proved that the virtual material method was effective in performing the modal analysis. The material strain energy method was another example to study efficient methods that could obtain the EM properties proposed by Ye et al. [16]. Mayer et al. [17] introduced segment-to-segment contact elements that established thick and thin layers of the joint interface of multi-degree-of-freedom systems. Masing’s experiment about the nonlinear response of bolted structure used the damping model and showed good agreement. Han et al. [18] identified the advantage of virtual material by comparing the spring-damping model and finite element method; the effectiveness was verified using the laser modal experiment. From the aforementioned references, it can be seen that determining EM’s physical properties is key in any modal analysis.

A modified fractal model (m-WK) [19] was proposed and verified in advanced. Thus, EM was used to simplify and analyze the joint interface of the bolted structure. The elastic modulus, the shear modulus, the Poisson’s ratio, and density of EM was calculated using the m-WK model. The finite element simulation and modal experiment were carried out to identify the reliability of EM. The results show that EM is reasonable and effective.

## 2. Properties of Equivalent Material (EM)

Figure 1 shows the typical schematic of the bolted structure. The contact was simplified as three smooth planes by replacing the rough contact interfaces with equivalent material. The properties of EM were calculated based on the actual surface profile, the contact area, contact stiffness, and other characteristics of the m-WK fractal model [19]. The adhesion, slide, and friction effect were ignored. The surface layer consisted of asperities and the substrate is assumed to be isotropic and the material was homogenous.

Figure 2 presents the schematic of oblique asperity contact, where $R_{e}$ is the equivalent asperity radius, α is contact angle, $\delta_{e}$ is the normal interference of the asperity, $\delta_{a}$ is the interference along the normal of contact point, $r_{a}$ is the tangential offset between the contact point and asperity summit, $a^{\prime }$ the truncated microcontact area, a is the real microcontact area, and $a=0.5a^{\prime }$. According to the limit position of $\delta_{e}$ and $R_{e}$, the range of contact angle is assumed to be 0≤α≤0.25π [19]. The nomenclatures of all parameters are listed in Nomenclature.

According to [9,15], when the contact angle is α=0, the elastic modulus e and the shear modulus $g_{e}$ of single asperity can be given as:(1)$$e=\frac{2E^{\prime }}{3\pi }\sqrt{\frac{R}{\delta }}$$
(2)$$g_{e}=\frac{16}{\pi }\sqrt[{3}]{1-\frac{\beta }{f}}G^{\prime }$$
where $E^{\prime }$ and $G^{\prime }$ are the equivalent elastic modulus and shear modulus, defined as $1/E^{\prime }=\left(1-\nu_{1}^{2}\right)/E_{1}+\left(1-\nu_{2}^{2}\right)/E_{2}$ and $1/G^{\prime }=\left(2-\nu_{1}\right)/G_{1}+\left(2-\nu_{2}\right)/G_{2}$. $E_{1}$, $G_{1}$, $\nu_{1}$ and $E_{2}$, $G_{2}$, $\nu_{2}$ are the elastic moduli, shear moduli, and Poisson’s ratios of two contact surfaces, respectively. β is the ratio between micro-slip tangential load and normal load. f is a constant coefficient of kinetic friction whose value is determined by the materials and the physical conditions of the interface, referred to as slip or micro-slip [9]. Kinetic friction is a surface effect that results from the asperity deformation and results from the energy lost through the bulk deformation [20]. R is the asperity radius and δ is the normal interference, which can be given as [13]:(3)$$R=\frac{{\left(a^{\prime }\right)}^{0.5D}}{2\pi G^{D-1}}$$
(4)$$\delta ={\left(G_{f}\right)}^{D-1}{\left(a^{\prime }\right)}^{1-0.5D}$$
where D is the fractal dimension (1<D<2) and $G_{f}$ is the fractal roughness parameter. R and δ are rewritten as $R_{e}$ and $\delta_{e}$ which can be given as [19]:(5)$$R_{e}=R{\left(1+\frac{r_{a}^{2}}{R^{2}}\right)}^{\frac{3}{2}}$$
(6)$$\delta_{e}=\delta_{a}\cos\alpha =\delta {\left(1+\frac{r_{a}^{2}}{R^{2}}\right)}^{\frac{3-2D}{D}}$$
where $\cos\alpha ={\left(1+\frac{r_{a}^{2}}{R^{2}}\right)}^{-0.5}$ and $1+\frac{r_{a}^{2}}{R^{2}}$ can be abbreviated as c for the specific contact angle. According to the limit position of $r_{a}$ and R, the range of α and c can be assumed as 0≤α<π/4 [19] and 1≤c<2. Substituting Equations (3)–(6) into Equation (1) results in:(7)$$e_{e}=\frac{2E^{\prime }}{3\pi }\sqrt{\frac{R_{e}}{\delta_{e}}}=\frac{\sqrt{2}E^{\prime }}{3\pi^{1.5}}{\left(G_{f}\right)}^{1-D}{a^{\prime }}^{0.5D-0.5}c^{\frac{5D-3}{4}}$$

The elastic modulus $E_{e}$ and the shear modulus $G_{e}$ of the EM can be calculated based on the size distribution density expression $n(a^{\prime })=0.5D\psi^{1-0.5D}{\left(a_{l}^{\prime }\right)}^{0.5D}{\left(a^{\prime }\right)}^{-1-0.5D},0<a^{\prime }\leq a_{l}^{\prime }$ as:(8)$$E_{e}=\int_{a_{ac}^{\prime }}^{a_{l}^{\prime }}\frac{e_{e}n(a^{\prime })a}{A_{r}}da^{\prime }=\frac{4\sqrt{2}E^{\prime }{\left(G_{f}\right)}^{1-D}\left(2-D\right)}{3\pi^{1.5}}{\left(a_{l}^{\prime }\right)}^{0.5D-1}c^{\frac{5D-3}{4}}\left[{\left(a_{l}^{\prime }\right)}^{0.5}-{\left(a_{ac}^{\prime }\right)}^{0.5}\right]$$
(9)$$G_{e}=\int_{a_{ac}^{\prime }}^{a_{l}^{\prime }}\frac{g_{e}n(a^{\prime })a}{A_{r}}da^{\prime }=\frac{16G^{\prime }}{\pi }\sqrt[{3}]{1-\frac{\beta }{f}}{\left(a_{l}^{\prime }\right)}^{0.5D-1}\left[{\left(a_{l}^{\prime }\right)}^{1-0.5D}-{\left(a_{ac}^{\prime }\right)}^{1-0.5D}\right]$$
where ψ is the domain extension factor (ψ>1), $A_{r}$ is the real area of the contact region, $a_{l}^{\prime }$ is the truncated area of the largest microcontact, and $a_{ac}^{\prime }$ is critical truncated area of oblique contact. $A_{r}$, $a_{l}^{\prime }$ and $a_{ac}^{\prime }$ can be given as [19]:(10)$$\frac{a_{l}^{\prime }}{A_{r}}=\frac{2\left(2-D\right)}{D\psi^{1-0.5D}}$$
(11)$$a_{ac}^{\prime }={\left(G_{f}\right)}^{2}{\left(\frac{H}{2E^{\prime }}\right)}^{2/(1-D)}{\left(\frac{\pi }{2}\right)}^{1/(1-D)}c^{\frac{3}{2(1-D)}}$$
where H is the hardness. It should be pointed out that the tangential force is generated for oblique asperity contact, which seems to be part of the shear load. However, the tangential force of a single asperity is transmitted to substrate. For the joint interface, the tangential forces almost internally cancel each other out, according to the distribution of their directions. Therefore, the shear modulus only takes the external load into consideration. The Poisson’s ratio $\nu_{e}$ and density $\rho_{e}$ of EM can be obtained as [9,15]:(12)$$\nu_{e}=\frac{E_{e}}{2G_{e}}-1=\frac{\sqrt{2}E^{\prime }{\left(G_{f}\right)}^{1-D}\left(2-D\right)c^{\frac{5D-3}{4}}\left[{\left(a_{l}^{\prime }\right)}^{0.5}-{\left(a_{ac}^{\prime }\right)}^{0.5}\right]}{96\pi^{0.5}G^{\prime }\sqrt[{3}]{1-\frac{\beta }{f}}\left[{\left(a_{l}^{\prime }\right)}^{1-0.5D}-{\left(a_{ac}^{\prime }\right)}^{1-0.5D}\right]}-1$$
(13)$$\rho_{e}=\frac{\rho_{1}h_{1}+\rho_{2}h_{2}}{h_{1}+h_{2}}$$
where $\rho_{1}$, $h_{1}$, and $\rho_{2}$, $h_{2}$ are the densities and thicknesses of two rough interfaces, respectively.

## 3. Experiments, Simulations, and Discussion

### 3.1. Samples Preparation and Measurment

In order to verify the accuracy the EM of the bolted structure, several sets of samples were fabricated for modal simulations and experiments. The size, material, machining method, surface roughness, and pre-tightening torque were selected as the variables to identify the characteristics of EM. Steel (1045) and aluminum alloy (AA 2024) were chosen as the sample materials. Table 1 gives their chemical compositions. Their material properties are shown in Table 2 [21,22].

The detailed values of the sample variables are listed in Table 3. Figure 3 exhibits photos of the specimens. Sample 2 (Figure 3b) is treated as the reference for the others. The other samples have different materials and geometrical properties.

Sample 1 has no interface (Figure 3a).The size of Sample 3 is smaller (Figure 3c).The material of Sample 4 is aluminum alloy (Figure 3d) while the others are steel.The interface of Sample 6 is finished by turning (Figure 3f) while the others are machined by milling (Figure 3e).

The interface profiles are measured using a three-coordinate measuring machine (ACCURA II AKTIV; Carl Zeiss AG, Oberkochen, Germany), as shown in Figure 4. The sampling interval was 50 μm and the sensor precision was 3 μm. Based on the measured data, the fractal parameters D and $G_{f}$ were calculated. However, there are many algorithms that calculate D and their computational accuracies have a great difference. Some results show that D obtained using the box-counting method are all approximately 1.1; the power spectral density method always makes a higher prediction. Meanwhile, the roughness-length method was regarded as the most reliable method for the actual self-similar machined surfaces. The structural function (SF) method, which is the same as the variogram method, was also relatively superior [23]. Furthermore, SF can extract D and $G_{f}$ at the same time without additional methods. Thus, SF was used to quantify the fractal parameters considering the computational accuracy and cost comprehensively. ψ was obtained from [13].

### 3.2. Finite Element Simulations

ANSYS [24,25] was selected as the finite element simulation software (ANSYS Workbench 14.0; ANSYS, Inc., Pittsburgh, PA, USA). To verify the effect of EM, the bolted structures with EM and without EM were both simulated. The predicted results strongly depended on the finite element mesh. For the simulations without EM, the three-dimensional structure model of the samples contained the rough profiles of the machined surfaces, which were fitted and created by Computer Aided Three-dimensional Interactive Application (CATIA V5; Dassault Systemes, Paris, France) [26]. Because of the irregular substrate, triangle mesh was applied; its minimum edge length was 25 μm. Figure 5a presents the 3D meshed model without EM, which includes 1,642,335 nodes. Since the number of layers had little influence on the global modal response [24], two layers were used and the amplified area showed the refinement view of the rough interface. For the simulations with EM, the contact solid was made up of three surfaces. The material of the intermediate solid was calculated using EM. The contact interface was smooth. Thus, computations can be performed on symmetric meshes with 463,880 nodes. Figure 5b shows the meshed model with EM.

The determination of EM was the key step of the simulations. According to the values of contact angle α, c can be given as c={$c_{i}$|1, 1.1, 1.2 … 1.8, 1.9}; their probability distribution density n(c) followed Gaussian distribution, which was evaluated as n(c)={$n(c_{i})$|0.2, 0.18, 0.16, 0.14, 0.1, 0.08, 0.06, 0.035, 0.025, 0.02}. Thus, the values of c can be given as:(14)$$c=\sum_{i=1}^{10}c_{i}n(c_{i})=1.2705$$

The tightening torque T need to be converted into the normal bolting force F for finite element simulation as:(15)$$F=\frac{T}{k_{m}d}$$
where d = 10 mm is the nominal diameter of the bolt and $k_{m}$ is a coefficient related to the bolt class. For steel hexagonal-headed bolt with a coarse pitch thread, $k_{m}$ is approximately equal to 0.2 when d = 10 mm. From [27], the contact force p between single asperity pair can be given as:(16)$$p=\frac{E^{\prime }\sqrt{\sigma_{p}/R}}{3.2}$$
where $\sigma_{p}$ is the standard deviation of surface heights The total force of the interface can be calculated as:(17)$$F=\int_{a_{ac}^{\prime }}^{a_{l}^{\prime }}pn(a^{\prime })da^{\prime }=\frac{E^{\prime }D\psi^{1-0.5D}\sqrt{\sigma_{p}}}{3.2\sqrt{2\pi {\left(G_{f}\right)}^{D-1}}}{\left(a_{l}^{\prime }\right)}^{0.5D}\left[{\left(a_{ac}^{\prime }\right)}^{-0.75}-{\left(a_{l}^{\prime }\right)}^{-0.75}\right]$$

Combined with Equations (11) and (15), $a_{l}^{\prime }$ can be obtained. β can be calculated as [9]:(18)$$\beta =\frac{p_{t}}{p_{n}}=\frac{p\sin\alpha }{p\cos\alpha }=\tan\alpha$$
according to the discrete values of c above, the values of β can be given as β = {$\beta_{i}$|0, 0.1, 0.2 … 0.8, 0.9}. Thus, $G_{e}$ and $\nu_{e}$ can be calculated by Equations (9) and (12) as:(19)$$G_{e}=\sum_{i=1}^{10}n\left(c_{i}\right)G_{e}\left(\beta_{i}\right),$$
(20)$$\nu_{e}=\sum_{i=1}^{10}n\left(c_{i}\right)\nu_{e}\left(\beta_{i}\right).$$

Based on Equations (8), (11), (13), (15), (19), (20), and other obtained middle variable values, all material properties of EM can be calculated as a list, shown in Table 4. Because the characteristics of the upper and lower interfaces are the same, density $\rho_{e}$ was equal to that of the sample, according to Equation (13). The tightening forces were imposed at the contact area between the bolts and the solid, as shown in Figure 6a. The bottom of the lower solid is fixed in accord with the experiment constraint condition.

The modal shapes of all samples were almost the same, as shown in Figure 7. Their resonant frequencies were discussed within the experiment results, in order to discriminate the reliability of EM. However, their difficulties in modeling and computation time of the finite element simulation were different, as shown in Table 5. The finite element models with EM had less nodes and elements, which are also displayed in Figure 5. Because of the surface profiles of the models with EM are smooth, the difficulty of modeling can be reduced without extra layer and refinement of mesh. Moreover, modeling with rough surface cost more than 10 times as much as modeling with EM. It indicated that EM can reduce computation time significantly to solve the problems with rough surfaces.

### 3.3. Modal Experiments

The classical frequency response method was applied to carry out the modal experiment whose principle is presented in Figure 8. The impact hammer was used to generate the excitation signal (channel A1) at different exciting points. The sensor signals (channels A2, A3, and A4) were transmitted to the computer through the signal acquisition card. Frequency response was estimated using the sensor signals from x, y, and z directions and the excitation signal and the estimation of z direction was the most remarkable and needed. Figure 9 shows the experiment field.

### 3.4. Results and Discussion

Figure 10 demonstrates the results of the modal experiments. It reveals the relationship between the resonant frequencies and the sample variables list in Table 3. Compared with the reference sample 2, size was the most sensitive variable that greatly increased the resonant frequency. Furthermore, the resonant frequency of reference sample 2 was 12,780 Hz. The result of sample 2 is 16,840 Hz which is 32% bigger. It indicates that interface should not be ignored for dynamic analyses. Fractal dimension D and fractal roughness $G_{f}$ of the interface were the key parameters. The turning surface with smoother surface obtained a higher resonant frequency. On the other hand, sample 7 made clear that the increment of the normal load can increase the contact stiffness of the interface and improve the modal characteristic. Softer material has smaller resonant frequency from the comparison between sample 2 and sample 4.

Table 6 presents the result comparisons between simulation and modal experiments of the bolted structure. Firstly, it can be seen that all experimental resonant frequencies were smaller than the simulation, which was mainly caused by the systematic error of the experimental process (such as deviations caused by sensors, machining, and assembly process) and the simulation calculation (such as deviations caused by measuring, fitting, and meshing process). Secondly, it was obvious that the errors between simulation results with EM and the experimental results were less than 10%; the minimum error reached 4.45%, while almost all errors of simulations without EM were greater than 10%. The modal prediction of bolted structure with EM has at least 4% higher computing precision.

In addition, there were important differences in the effectiveness among different EM variances compared with the reference sample 2. Sample 3 explained that the smaller size obtained smaller error because the area of rough interface was smaller. Moreover, samples 4 and 5 had relative greater errors. These illustrated that the material of the bolted structure and the surface roughness and fractal parameters of the interfaces had significant influence on the EM and modal response. On the other hand, sample 6 showed that the machining method was not vital for the EM process, as long as the interface characteristics were guaranteed. Furthermore, the increment of tightening torque would increase the contact stiffness and reduce the damping effect of the joint interface.

## 4. Conclusions

The equivalent material (EM) of the joint interface was used to analyze the modal performance of the bolted structure. Contact angle α was introduced to calculate the elastic modulus, the shear modulus, the Poisson’s ratio, and density of EM using a modified fractal model based on oblique asperity contact.

From the frequency response experiment, it was revealed that the interface should not be ignored for dynamic analyses. Dimension, material, machining method, surface roughness, and pre-tightening torque of the interface had a greater influence on the resonant frequency. Small bolted structures with a smoother interface and bigger tightening torque via the turning method obtained a higher resonant frequency.

The Finite element method was applied to discriminate the resonant frequencies of the bolted structures with and without EM. The material properties of EM were calculated using experiment conditions. The simulation results were compared with the classical frequency response experiment. The errors between experiments and the bolted structures with EM were less than 10%, which were much less than those without EM, while almost all errors of simulations without EM were greater than 10%. The modal prediction of the bolted structure with EM had at least a 4% higher computing precision. Furthermore, EM significantly reduced computation time to solve the problems with rough surfaces.

It can be concluded that the bolted structures with EM were reliable and reasonable. Moreover, the EM method can improve the accuracy and reduce the computation cost. This method can be extended to other mechanical structures that consider dynamic characteristics of the machined rough interfaces.

## Figures and Tables

**Figure 1 materials-12-03004-f001:**
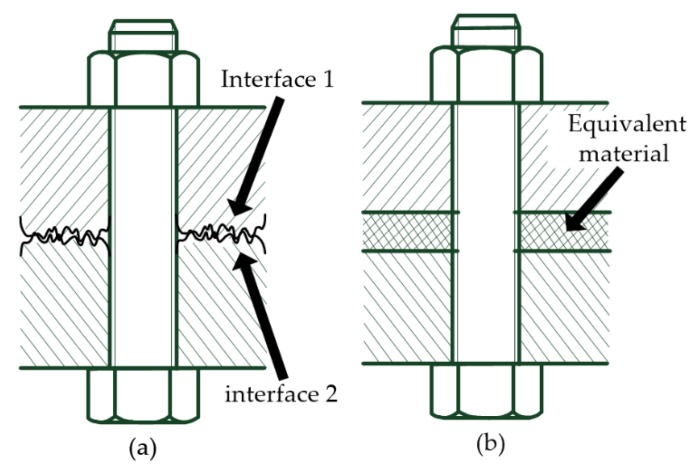
Schematics of bolted structure with joint interfaces of: (**a**) actual topography; (**b**) equivalent material (EM).

**Figure 2 materials-12-03004-f002:**
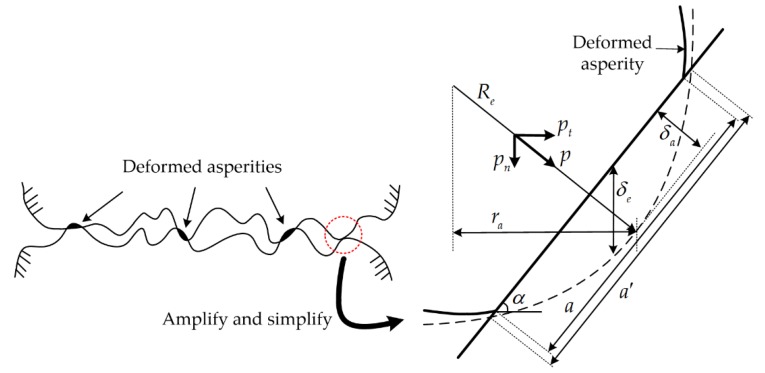
Schematic of oblique asperity contact.

**Figure 3 materials-12-03004-f003:**
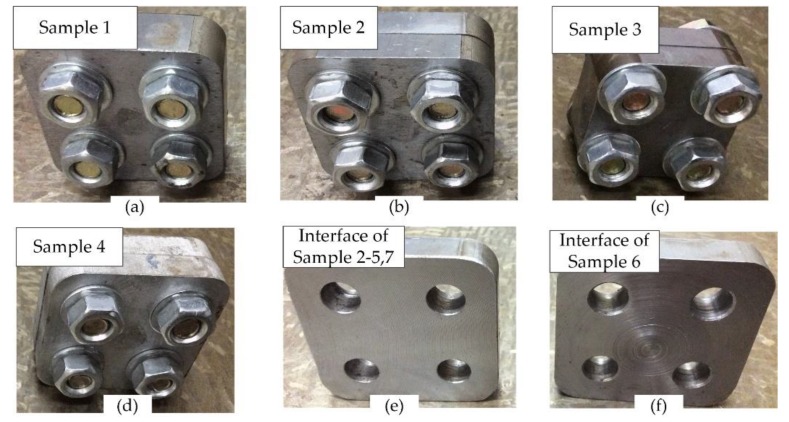
Some photos of different samples and interfaces. (**a**) Sample 1; (**b**) Sample 2; (**c**) Sample 3; (**d**) Sample 4; (**e**) interface of Sample 2–5, 7; (**f**) interface of Sample 6.

**Figure 4 materials-12-03004-f004:**
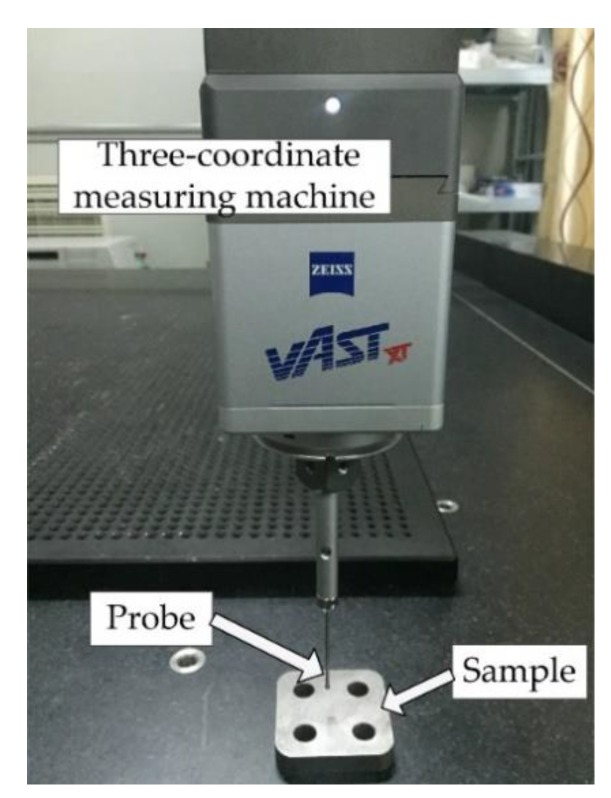
Measuring the interface profile of the sample using the three-coordinate measuring machine.

**Figure 5 materials-12-03004-f005:**
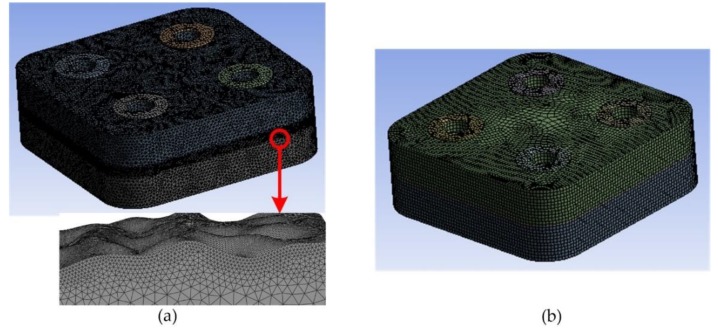
3D mesh of finite element simulation. (**a**) the meshed model without EM and a view to the amplified area; (**b**) the meshed model with EM.

**Figure 6 materials-12-03004-f006:**
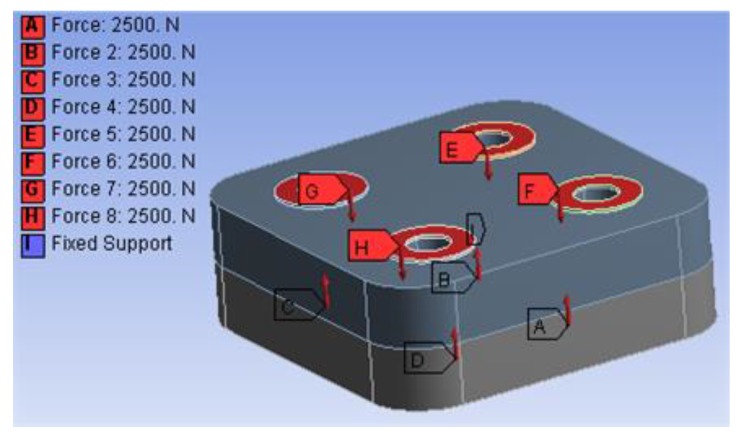
Loads and constraint for bolted structure.

**Figure 7 materials-12-03004-f007:**
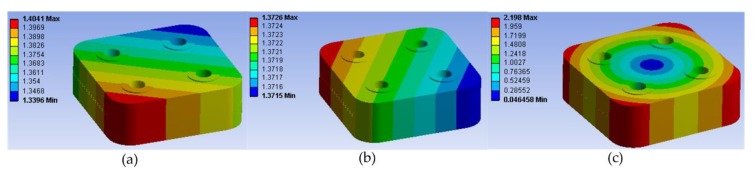
Modal shapes of different orders: (**a**) First order; (**b**) second order; (**c**) third order.

**Figure 8 materials-12-03004-f008:**
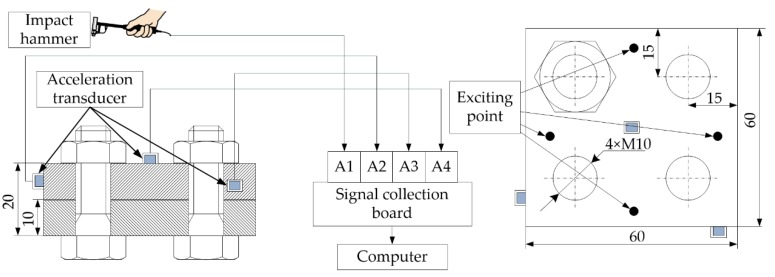
The principle of the modal experiment.

**Figure 9 materials-12-03004-f009:**
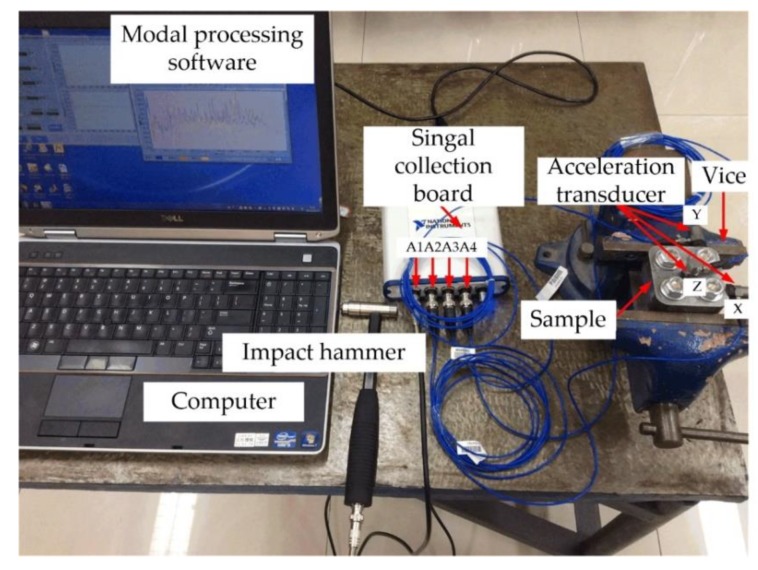
The field of the modal experiment.

**Figure 10 materials-12-03004-f010:**
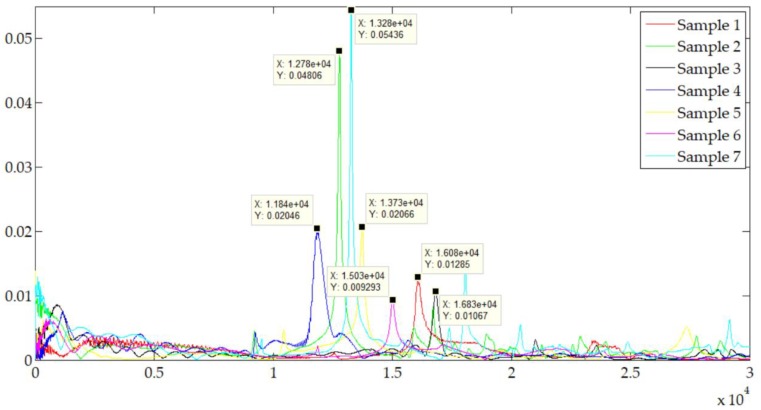
The results of the modal experiments.

**Table 1 materials-12-03004-t001:** Chemical compositions of steel 1045 and aluminum alloy 2024.

Material	Chemical Compositions
1045	Fe balance; Mn 0.5–0.8; C 0.42–0.5; Cr ≤ 0.3; Ni ≤ 0.25; Cu ≤ 0.25; Si 0.17–0.37
AA 2024	Al balance; Cu 3.7–4.5; Mg 1.2–1.5; Zn 0.25; Fe 0.2; Mn 0.15–0.8; Si 0.15; Ti 0.15

**Table 2 materials-12-03004-t002:** Material properties of steel 1045 and aluminum alloy 2024.

Material	Density, ρ (g/cm^3^)	Elastic Modulus, E (GPa)	Shear Modulus, G (GPa)	Poisson’s Ratio, ν	Coefficient of Kinetic Friction, f	Hardness, H (HV)
1045	7.85	180.0	69.2	0.30	0.15	255
AA 2024	2.78	73.2	27.5	0.33	0.17	126

**Table 3 materials-12-03004-t003:** Values of the sample variables.

Sample	Size (mm^3^)	Material	*D*	*ψ*	*G_f_* (mm)	Machining Method	Tightening Torque, *T* (N·m)
1	60 × 60 × 20	1045					5
2	60 × 60 × 10	1045	1.17	2.3411	3.7 × 10^−5^	milling	5
3	50 × 50 × 10	1045	1.16	2.3549	9.5 × 10^−5^	milling	5
4	60 × 60 × 10	2024	1.18	2.3276	1.2 × 10^−5^	milling	5
5	60 × 60 × 10	1045	1.09	2.4593	5.9 × 10^−4^	milling	5
6	60 × 60 × 10	1045	1.16	2.3549	8.1 × 10^−5^	turning	5
7	60 × 60 × 10	1045	1.17	2.3411	4.1 × 10^−5^	milling	10

**Table 4 materials-12-03004-t004:** Material properties of EM for finite element simulation.

Sample	Elastic Modulus, $E_{e}$ (GPa)	Shear Modulus, $G_{e}$ (GPa)	Poisson’s Ratio, $\nu_{e}$
2	53.2	25.9	0.22
3	39.5	26.2	0.16
4	29.6	10.4	0.30
5	20.0	28.0	0.08
6	42.9	26.2	0.17
7	52.3	25.9	0.21

**Table 5 materials-12-03004-t005:** Statistics of different finite element modeling methods.

Modeling Methods	Mean Number of Nodes	Mean Number of Elements	Computation Time (Minute)
with EM	463880	304716	2.5
without EM	1642335	1129607	32

**Table 6 materials-12-03004-t006:** Comparisons of resonant frequency between simulations and modal experiments.

Sample	Experiment ×10^4^ (Hz)	Simulation without EM ×10^4^ (Hz)	Error %	Simulation with EM ×10^4^ (Hz)	Error %
1	1.608	1.673	4.04	1.673	4.04
2	1.278	1.414	10.64	1.36	6.42
3	1.684	1.845	9.56	1.759	4.45
4	1.184	1.397	17.99	1.297	9.54
5	1.373	1.578	14.93	1.508	9.83
6	1.501	1.709	13.86	1.583	5.46
7	1.328	1.531	15.29	1.404	5.72

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
