# Peer review of "Modal Analysis of Bolted Structure Based on Equivalent Material of Joint Interface"

_materials, 2019, doi:10.3390/ma12183004_

Round 1

Reviewer 1 Report

Dear Authors,

The paper is worth for publication. As a reviewer, I have minor comments to the paper - see below:
Page 3 line 82 – please use plural form of “reference”
Page 4 line 112 – “The fractal parameters D and G are calculated by structural function method” – Please provide more details. It is extremely important to compare obtained results of fractal dimension. You can use some papers which discussed different methods of fractal analysis – regarding surface:
Kotowski, P. (2006). a) Fractal dimension of metallic fracture surface. International journal of fracture141(1-2), 269-286
Carpinteri, A., & Mainardi, F. (Eds.). (2014). Fractals and fractional calculus in continuum mechanics(Vol. 378). Springer.
The caption should be changed – please rewrite this caption
Can you discuss the influence of boundary condition (please show it!) on the obtained results?

Reviewer 2 Report

In my opinion authors must correct/clarify the following issues:

Line 34. Please use “and” instead of “And”. Please do the same in lines 37, 44, 52, 60, 142…

Line 44. For the sake of clarity, please define all the acronyms appearing in the main text.

Line 90. Please provide a reference for the range of α

Line 113. Please use the format included in the journal template for the table headings. Do not place it below the table. In addition, for the sake of clarity, be more precise, include in the main text the chemical composition of the steel and aluminum used in the study including the AISI code and include it in the table 1. Besides a table including the main mechanical properties is needed.

Line 119. Please use “ANSYS” instead of “ANAYSIS”

Line 120. The FE simulation is not properly described since just a general view of one mesh (Figure 5) and just a reference to the FE software (ANSYS) are included in the manuscript. Authors must include a deep description of the FE modeling from the CATIA model, including mesh convergence, type of elements, number of nodes, mesh size at the contact zones, how the mesh is generated? using automesh tool?, boundary conditions, material properties, type of analysis and so on. In addition authors must show how the equivalent material is included in FE simulations. The mesh in the simulations including equivalent material zone is the same that the one used in sumulations without equivalent material zone?. The distribution and element sizes are the same for both cases?.

Line 127. Please use italics for variables included in the main text.

Line 136. Please use the journal template for the format of the table headings.

Line 136. Table 2. The heading of the third and fifth columns are the same. I guess the values included in the fifth column corresponds to the case using equivalent material. Please correct it. Anyway, authors must describe deeply how these values are obtained from the FE simulations, and they should include FE images of the main vibrations modes given by simulations for each case of study.  

Line 156-157. For the sake of clarity, please explain deeply how this is done and include it in the manuscript.

Line 161. This conclusion is not properly discussed in the paper, since only the values of the error considering one case for each is included, please discuss this deeply in the manuscript.

Line 205. Please use “Zhang M.;” instead of “Zhang M.,” in reference 18. “Han R.; Li G.; Gong J.; Zhang M., Zhang K.”

Reviewer 3 Report

The article contains interesting research results, but it is edited incorrectly and the methodology is not described completely. The data provided is not sufficient to check that investigations were conducted correctly. So it requires a thorough improvement to meet the standards of the journal Materials.

line 44. The abbreviations GW, BGT etc must be explained.

Lines 62, 63: "is used to is used to" ?

line 72: Authors says: "The surface is assumed to be isotropic ..." In mathematics, "surface" has no volume. So, it does not exhibit physical properties as isotropy or homogeneity. Did you mean "layer"?

Eg. (2): What is the ge parameter in this equation.

In line 70 the Authors says that the friction in EM is ignored. However, to determine the shearmodulus Ge (Eq. (9)) and Poisson ratio (Eq. (10)) a friction coefficient is necessary. Please explain this situation to the readers.

According to the caption Table 1 presents PROPERTIES of the samples. Table presents views, not PROPERTIES of the samples!

What is the clear difference in the photos presented in figs. 3a, 3b, 3c and 3e named as "aluminium", "50x50x10", reference sample" etc.?????

Chapter 2 consists the general description of the EM based on the data from the references cited. The authors should give information sufficient enough to reproduce the experiment/investigation.

The determination methods and values of physical-mechanical properties of the materials which has been used to determine EM (Eqs. (1)-(11) must be included in this paper. The data provided should allow to apply the EM by other researchers.

Chapter 3 is too general. It should be divided into the sections describing separately the experimental procedure, the  numerical model description and the results of both with discussion.

Fig. 5. "The mesh grid" ?

Besides showing the view of mesh, the Authors did not describe the numerical model in a sufficient way. There are a lack of (i) description of material properties of bolted structure, (ii) boundary conditions, (iii) implementation of EM, (iv) results of mesh sensitivity analysis, etc.

The results presented in figure 8 has not been discussed. The Authors stated laconically only in lines 128 and 129 that "Figure 8 demonstrates the results of the modal 128 experiments."

The caption for the tables must appear above the table.

For each sample, table 2 contains two values of resonant frequency determined by FEM. Why?

The terminology and English grammar requires thorough improvement.

Reviewer 4 Report

Memorandum

Subject: Review, August 16, 2019

Title:   Modal analysis of bolted structure based on equivalent material of joint interface Kai

Zhang 1, *, Guoxi Li1, Jingzhong Gong2 and Fei Wan1

 1 College of intelligence science and technology, National University of Defense Technology, No.109 Deya

    Street, Changsha 410073, P. R. China; [email protected] (G.L.); [email protected] (F.W.)

 2 School of Mechanical Engineering, Hunan International Economics University, No.822 Fenglin Street,

    Changsha 410073, P. R. China; [email protected] (J.G.)

 * Correspondence: [email protected]; Tel.: +86-155-8084-5984 (K.Z.)

Comments:

The authors should add a nomenclature to identify all the parameters and abbreviations used throughout the paper. On line 118, the name ANSYS is misspelled, also, CATIA and ANSYS have to be referenced. The finite element model shown in Figure 5 lacks information, the authors should include type of element used, number of element and nodes. Also, the type of analysis and boundary conditions should be shown. The authors should show more than one view of figure 5 to allow good exposure of the model and be able to note the details. The finite element or the simulation description of this work lacks good explanation, it is not fully outlined in the paper. A more expressive representation of how it was implemented is very helpful The authors mention Finite element simulation and the results showed the simulation tabulation format, it would serve the reader better if they include a comparison plot showing the variation. Additionally, the authors should include a fringe plot or more showing the modal response under different conditions. This will give more representative description of how the simulation is performing. The data shown in figure 8 is not explained, no discussion exist in the text highlighting the observations in the plot. A clarification is recommended. The conclusion can be improved, as is, it does not deliver a clear message. The authors should consider a bullet format type statements citing the findings and any issues that may have impacted the outcome of the data obtained.

Round 2

Author Response

Mistakes are corrected in the entire manuscript. English language and style are improved.

Reviewer 3 Report

The resubmitted manuscript is clearer and structured significantly better than the initially submitted one. However, some issues related to the use of technical terms and grammar were still observed. Below a detailed description of the observed issues is provided:

Nomenclature: νe is "ratio" not "ratios"

"Poisson" should be "Poisson's". This applies to the entire manuscript.

Considering assumed style of nomenclature, what is the unit of "contact angle"?

According to section Nomenclature, a is the elastic microcontact area. However, in line 82 the Authors says: a is the real microcontact area. Please check that all variables are suitably defined in the text. This applies to the entire manuscript. Moreover the meaning of a' parameter is different in the text and section Nomenclature.

line 86: Why all the variables and parameters used in the text are not included in the section Nomenclature? FOR EXAMPLE, the shear modulus Ge, tightening moment T, contact pressure p, etc.,  are not included in this section.

line 90: Sentence: "E2, G2, ν2 are the elastic moduli, shear e moduli and Poisson ratios of two contact surfaces, respectively". should be "E2, G2, ν2 are the elastic modulus, shear modulus and Poisson's ratio of two contact surfaces, respectively."

Equation (1): Meaning of δ parameter should be included in the text.

Line 110: “and of EM” ?

line 112: "thickness" should be replaced by "thicknesses", as you used the plural of the noun "density".

line 118: The abbreviation of steel as AA1045 is not correct! Abbreviation "AA" is allocated for aluminium or aluminium alloys. I suggest to use "1045" or more formal "AISI 1045".

The entire manuscript: Why word steel is written with a capital letter?

The entire manuscript: There is collision between denotation of shear modulus G (Table 2) and fractal roughness parameter G (section Nomenclature).

Line 122. I suggest to use "properties" or "material and geometrical properties" instead of "variables".

Line 138: The sentence "The fd3 program method is effective but rely on the microscopic magnification and resolution of the data [23]." is not essentially connected with the proceeed sentence and the whole paragraph. It looks like digression from other article. Please make this paragraph clearer. It is also confused that fd3 means program or method.

The entire manuscript: The Tables and Figures should appear in the manuscript after they were  called out in the text.

line 165: [] ?

Equation (15): It is clearer to define F as T/km·d, with explanation of value of coefficient km = 0.2.

The value of coefficient km mainly depends on the tightening moment value and bolt class. Equation (15) in the revised version of manuscript suggests that the T is always divided by 0.2d.

In lines 168-169 the Authors says “From reference [27], the contact force between single asperity pair can be given as: …”. In the reference [27] there is no information how to evaluate the contact force based on the Eq. (16).

line 189: "Modeling" should be replaced by "modeling".
